# Odor-active compounds from the gonads of *Mesocentrotus nudus* sea urchins fed *Saccharina japonica* kelp

**Satomi Takagi**[1], **Yoichi Sato**[2,3], **Atsuko Kokubun**[4], **Eri Inomata**[1], **Yukio Agatsuma**[1]*

**1** Laboratory of Marine Plant Ecology, Graduate School of Agricultural Science, Tohoku University, Aza-Aoba, Aramaki, Aoba, Sendai, Miyagi, Japan, **2** Riken Food Co., Ltd., Miyauchi, Tagajyo, Miyagi, Japan, **3** Nishina Center for Accelerator-Based Science, RIKEN, Hirosawa, Wako, Saitama, Japan, **4** Food Analysis Laboratory, Quality Assurance Division, RIKEN VITAMIN Co., Ltd., Aoyagi, Soka, Saitama, Japan

* yukio.agatsuma.c7@tohoku.ac.jp

**Data Availability Statement:** All relevant data are within the paper and its Supporting Information files.

## Abstract

Gonad size, color, texture and taste of *Mesocentrotus nudus* sea urchins collected from a barren can be improved by a short-term cage culture while being fed fresh *Saccharina japonica* kelp during May–July. We investigated the effect of *S. japonica* feeding during May–July on the improvement of gonad flavor in *M. nudus* collected from a barren. After feeding, we analyzed the odor-active volatile organic compounds (VOCs) from the gonads using gas chromatography-mass spectrometry and GC-sniffing analyses and compared to those from the gonads of wild sea urchins from an *Eisenia* kelp bed (fishing ground) and a barren. A total of 48 VOCs were detected from the gonads of cultured and wild sea urchins. Of them, a larger number of odor-active compounds were detected in the gonads of cultured sea urchins (25) than in those from the *Eisenia* bed (14) and the barren (6). Dimethyl sulfide from the gonads of sea urchins from the barren was described as having a strong, putrid odor. Sea urchin-like aromas were attributed to 2-butanol, 2-ethylhexanol, benzaldehyde and ethylbenzene from the gonads of cultured sea urchin and those of the *Eisenia* bed. Kelp feeding decreased the putrid odor from dimethyl sulfide, and enhanced pleasant, sweet aromas.

## Introduction

Olfaction plays a dominant role in the taste of food [1]. In the human brain, the perception of flavor occurs in the neural system, which affects food preference and appetite [2]. Yamamoto et al. [3] demonstrated that the odor of the fragrant tea olive *Osmanthus fragrans* decreases appetite and food intake, resulting in a decrease in body weight. Furthermore, odor can change the taste of food [4].

Sea urchin gonads are a highly valued cuisine and, particularly, a favorite sushi ingredient in Japan [5]. Japan is the largest consumer of sea urchin gonads in the world, accounting for approximately 90% of the total world sea urchin landing [6]. Past studies revealed that gonads with large size, bright orange or yellow color, medium hardness, high sweet-tasting alanine and low bitter-tasting arginine contents are preferred [7–9]. The factors concerned with gonad quality have been studied [10–13]. Flavor is an important factor in the sensory quality of sea

**Funding:** This study was financially supported by the Grand-in-Aid for JSPS Fellow [grant number 17J02308] from Japan Society for the Promotion of Science [S.T.], and basic science research funding of Riken Food Co., Ltd. [Y.S.] and RIKEN VITAMIN Co., Ltd. [A.K.]. The funders provided financial support in the form of salaries for authors [S.T., Y.S., A.K.] and research materials, but did not have any additional role in the study design, data collection and analysis, decision for publication, or preparation of the manuscript.

**Competing interests:** Authors [Y.S.] and [A.K.] received salary from Riken Food Co., Ltd. and RIKEN VITAMIN CO., Ltd, respectively. There are no relevant patents or marketed products to declare. This does not alter our adherence to PLOS ONE policies on sharing data and materials.

urchin gonads [10, 14]. Some components associated with the flavor differ among sea urchin species [15–18]. In *Evechinus chloroticus*, flavor and the associated volatile compounds in gonads vary between the north and south of New Zealand [15, 16]. To date, the odor-active volatile compounds in the gonads of *E. chloroticus* [15, 16], *Paracentrotus lividus* [17] and *Mesocentrotus nudus* [18] have been identified. Some of them were inferred to be associated with sea urchin aroma [17, 18], but the compounds with the "sea urchin-like" aroma have still not been determined.

*Mesocentrotus nudus* accounts for more than two-thirds of the total sea urchin landing in Japan together with *Strongylocentrotus intermedius* and is the most expensive sea urchin gonad in the world [19]. This species densely distributes on barrens [20], where the gonads show small size [21] and deteriorated color, texture, and taste [13, 22], therefore no commercial value [20]. Recently, research on the improvement of the gonad quality of *M. nudus* from barrens by short-term culture techniques has been concentrated on in northern Japan [13, 23]. Takagi et al. [8] first improved the whole gonad quality (size, color, texture and taste) of barren individuals to a more desirable level compared to that of individuals from an *Eisenia* kelp bed (fishing ground) by feeding them fresh cultivated *Saccharina japonica* during May–July. They evaluated that the gonad smell of sea urchins fed *S. japonica* and that of those collected from an *Eisenia* kelp bed were more desirable than that of those from a barren, as determined by a panel test [8]. The results suggested that the odor-active compounds in gonads could vary in accordance with their habitats.

In the present study, *M. nudus* were collected from a barren and cultured in cages suspended offshore and fed fresh cultivated *S. japonica* during May–July. After feeding, the odor-active volatile compounds of the gonads were analyzed by gas chromatography-mass spectrometry (GC-MS) and GC-sniffing methods and compared to those from the urchin gonads collected from an *Eisenia* kelp bed and a barren. Takagi et al. [8] conducted a sensory evaluation and measurements and analyses of gonad size, developmental stages, color, texture and free amino acid contents of sea urchins cultured and collected from the same algal beds by the same methods on the same day with the present study. The purpose of this study is to (1) clarify the effect of *S. japonica* feeding to improve gonad flavor of sea urchins from a barren and (2) identify the compounds with sea urchin like aroma.

## Materials and method

### Ethics statement

*Mesocentrotus nudus* were collected from a site in the Shizugawa Bay, Miyagi Prefecture, Japan that is not privately-owned or protected in any way. Field studies did not include endangered or protected species. All experimental procedures on animals were in compliance with the guidelines of Miyagi Fisheries Cooperative Association and Miyagi Prefectural Government.

### Sea urchin samples

A total of 500 adult *M. nudus* (46–55 mm diameter) were collected by scuba divers from a barren at depths of 2.5–3.0 m off Nojima Island, Shizugawa Bay, Miyagi Prefecture (38°40´N, 141°30´E) on 6 May 2016. Immediately after collection, 100 sea urchins were placed in each of five cages suspended horizontally along a straight line at a depth of approximately 4.5 m at a wave-sheltered site off Areshima Island (38°40´N, 141°27´E) in the bay until 19 July 2016. The cages were cuboid (90 × 87 × 45 cm) with 3 cm mesh and made of polyethylene. The sea urchins were kept without food for 5 days until the start of the experiment. Fresh *S. japonica* cultivated were fed to sea urchins *ad libitum* every 7–10 days for 63 days from 11 May to 13 July. On 19 July, 20 cultured sea urchins were collected randomly from the cages, and 20 wild

sea urchins were collected from each of the barren where we collected the sea urchins for the cage culture and an *Eisenia bicyclis* kelp bed at depths of 2.2–3.0 m off Nojima Island. Thus, three different groups, cultured sea urchins (CSU), and wild sea urchins collected from a barren (BSU) and an *E. bicyclis* bed (ESU), were designed. After collection, the sea urchins were kept in two cool boxes with moist urethane mats immersed in seawater and then transported to the factory of Riken Food Co. Ltd. in Tagajo (38˚16´N, 141˚0´E). These sea urchins were dissected, and the gonads were isolated and soaked 3 times in 5˚C sterile seawater. Thereafter, they were patted dry on bleached cotton at 4˚C for 30 minutes according to Kinoshita et al. [24]. No gamete released from the gonads was observed. A total of ca. 5 g gonads was randomly collected from one group of gonads and stored in a polystyrene storage container at 4˚C until further analyses. Three containers were prepared for each group (n = 3). Headspace sampling, GC-MS analysis and GC-sniffing analysis were conducted according to Sato et al. [18] within 48 hours after dissection. Takagi et al. [8] showed no significant differences in test diameter among these sea urchins (CSU, ESU and BSU), and almost the gonads were in the growing stage.

## Large volume static headspace sampling

For analysis of volatile organic compounds (VOCs) of sea urchin gonads, headspace volatile compounds were collected in a large volume static headspace (LVSH) system (Entech 7100A series, Entech Instruments Inc., Simi Valley, CA, USA). Sea urchin gonads were analyzed within 48 hours after collection. From each container, ca. 5 g gonads were sealed in a 375 ml glass jar for measurement via LVSH, which was stored in an incubator (DK400, Yamato Scientific Co., Ltd., Tokyo, Japan) at 30˚C for 10 min. After incubation, 150 ml of headspace gas was vacuum-extracted from the glass jar. The VOCs were desorbed by thermodesorption using a preconcentrator (Entech 7100A series, Entech Instruments Inc.) and injected into the GC-MS system.

## GC-MS analysis

Quantification of the VOCs was performed using an Agilent 6890 series gas chromatograph (Agilent Technologies Inc., Palo Alto, CA, USA) equipped with an Agilent 5975B mass-selective detector and a sniffing port. One half of the column flow was directed to the MS system, while the other half was directed to the heated sniffing port. The GC-MS system was equipped with a DB-WAX column (60 m × 0.25 mm i.d., 0.5 μm film thickness; 122–7063; Agilent Technologies Inc.). The GC injector temperature was 250˚C. Analyses were carried out using helium as the carrier gas at an average flow rate of 27 cm sec$^{-1}$ with the following temperature program: 40˚C for 5 min, an increase at 5˚C min$^{-1}$ to 240˚C, followed by a final 5 min hold at 240˚C.

Mass spectrometry was carried out in scan mode using an electron ionization voltage of 70 eV and a scan range from m/z 10 to 300 with a scan rate of 1.58 scans per sec. Analysis of VOCs was performed using Powered Pro software (Wiley 11N17main, Agilent Technologies Inc.). Each VOC was identified by similarity search [25] using the library of the software (Wiley 11N17main, Agilent Technologies Inc.). When a VOC was detected in triplicate analysis, this determined the presence of the VOC in the group. The VOCs detected in the present study were compared to those of Sato et al. [18]. The relative amounts of each VOC detected by GC-sniffing analysis were calculated based on the peak areas in the chromatograms.

## GC-sniffing analysis

GC-sniffing analysis was conducted for each group. One half of the column flow was directed to a heated sniffing port (ODP2 olfactory detection port, Gerstel GmbH & Co. KG, Mülheim an der Ruhr, Germany). To the sniffing port, humidified air (50–75% relative humidity) was carried at 1.02 ml min$^{-1}$ to prevent the nose from drying out. The three panelists, who are well

versed in sea urchin gonad quality and share common perceptions, recorded the retention time and the related description of the aroma compounds by writing in a paper. They could record without taking off their nose from the sniffing port.

## Statistical analyses

Statistical analyses were conducted using JMP 10 software (SAS Institute Inc., Cary, NC, USA). Differences in the peak areas of compounds detected by GC-MS analysis among the three groups were analyzed with one-way analysis of variance (ANOVA). The values of not detected compounds were inputted as 0.0001. Tukey's multiple comparison test was performed as a *post hoc* test.

# Results

Typical total ion chromatograms of the VOCs from sea urchin gonads of each group are shown in Fig 1. The VOCs we detected are shown in Table 1 and compared to those reported by Sato et al. [18]. A total of 48 VOCs were detected in the gonads of CSU (48), ESU (48) and BSU (46). Of them, S-methyl thioacetate and bromoform were not detected in the gonads of BSU. These compounds could be categorized into the following chemical families: alcohols (7), aldehydes (8), aromatic compounds (9), esters (4), halomethanes (4), hydrocarbons (7), ketones (6), and others (3; dimethyl sulfide (DMS), acetonitrile, and bis-(methylthio)-methane). Of them, 35 compounds were also detected by Sato et al. [18]. The peak areas of the odor-active VOCs detected by GC-sniffing analysis are shown in Table 2. Odors of 25, 14, and 6 compounds detected from the gonads of CSU, ESU and BSU, respectively, were described.

## Alcohols

Seven alcohols, methanol, 2-propanol, ethanol, 2-butanol, propanol, butanol and 2-ethyhexanol, were detected. The odors of four alcohols were described. There were no significant differences in the peak areas of each odor-active alcohol among each group. The odors of 2-butanol from the gonads of CSU and ESU were described as acceptable seafood and hey smells and as an unpleasant sea urchin-like odor, respectively. 2-Ethylhexanol from the gonads of CSU and ESU was described as having a sea urchin-like aroma. 2-Propanol from the gonads of CSU was described as having kelp and hey smells.

## Aldehydes

Eight aldehydes, acetaldehyde, dimethoxymethane, acetal, 2-ethylhexanal, octanal, nonanal, decanal and benzaldehyde, were identified from the sea urchin gonads of each group. The odors of all aldehydes could be described. There were no significant differences in the peak areas of each aldehyde among the groups. Acetaldehyde, dimethoxymethane and octanal from gonads of ESU were described as having a putrid odor. Acetal and 2-ethylhexanal from the gonads of CSU were described as having a sweet aroma. Octanal from the gonads of CSU and BSU were described as having green scents. Benzaldehyde from the gonads of CSU was described as having a sea urchin-like aroma.

## Aromatic compounds

Nine aromatic compounds, methylcyclohexane, benzene, toluene, ethylbenzene, xylene, ethyltoluene, trimethylbenzene, styrene and dichlorobenzene, were detected from the gonads of sea urchins of each group. The odors of seven compounds were described. Except for styrene, there were no significant differences in aromatic compound peak areas among each group.

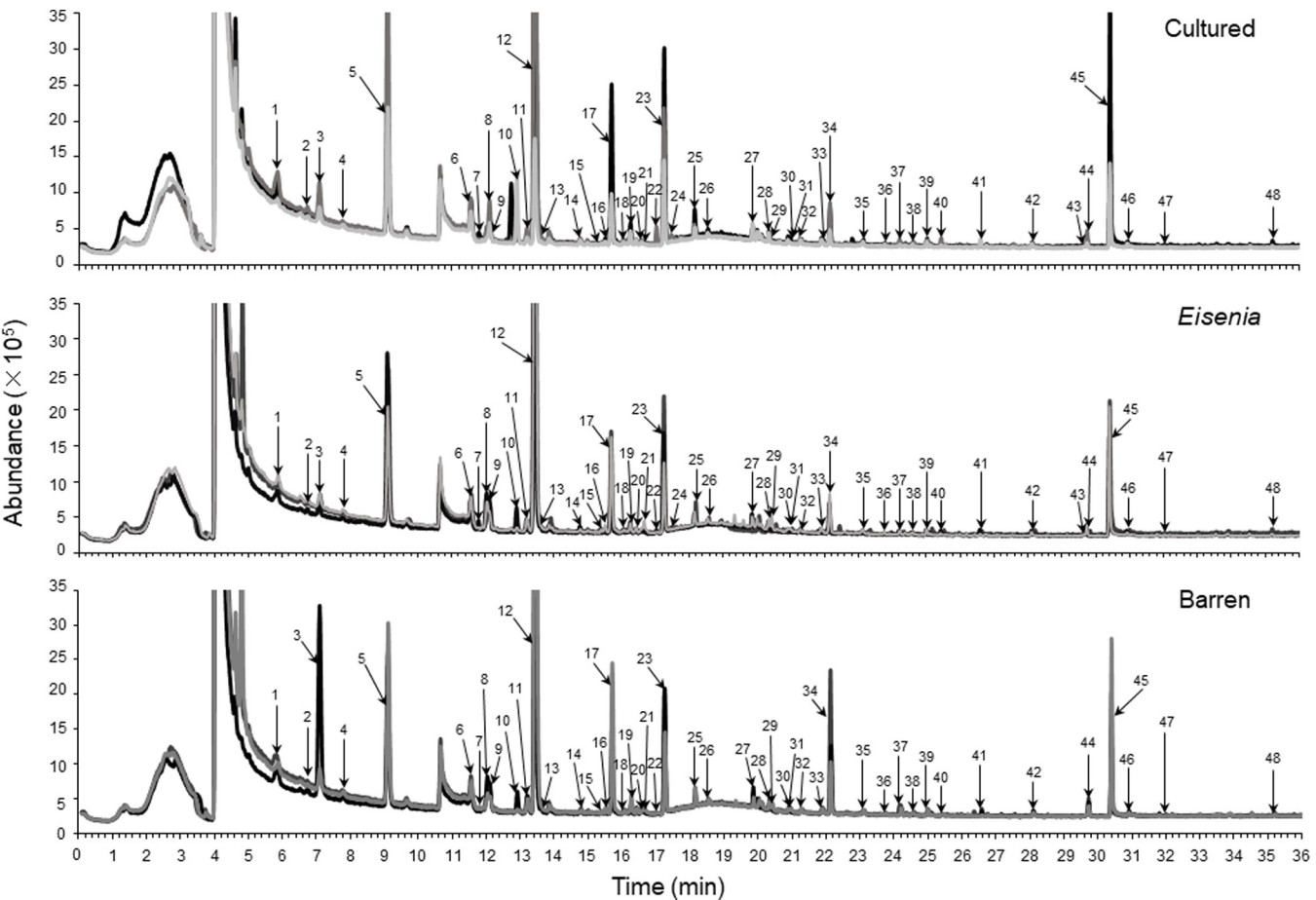

**Fig 1. Chromatograms of volatile organic compounds (VOCs) in the gonads of *Mesocentrotus nudus* using gas chromatography-mass spectrometry (GC-MS) (N = 3).** Cultured, *Eisenia* and Barren indicate cultured sea urchins, and sea urchins collected from an *Eisenia bicyclis* bed and a barren, respectively. Compounds were identified by peak numbers, as shown in Table 1.

Benzene from the gonads of CSU was described as having a heavy note, although that from ESU was described as having a mint aroma. Toluene from the gonads of CSU and ESU were described as having fruit-like and sweet aromas, respectively. The peak area of toluene from the gonads of CSU was large compared to that of the other groups. Ethylbenzene from the gonads of CSU was described as having sea urchin and baked fish-like aromas compared to the noted putrid fish and sour odors from that of ethylbenzene from ESU gonads. The peak area of styrene from the gonads of BSU was significantly larger than in the other groups ($P < 0.001$). Styrene from the gonads of CSU and ESU was described as having a hey smell.

## Esters

Four esters, ethyl acetate, propyl acetate, S-methyl thioacetate and butyl acetate, were identified. S-methyl thioacetate was not detected in the gonads of BSU. The odors of all detected esters were described. Except for S-methyl thioacetate, there were no significant differences in peak areas among each group. Ethyl acetate and propyl acetate from the gonads of CSU were described as having sweet aromas. S-methyl thioacetate from the gonads of CSU showed a markedly larger peak area than those of other groups and was described as having a heavy note

**Table 1. Volatile organic compounds (VOCs) detected from gonads of *Mesocentrotus nudus* using gas chromatography-mass spectrometry (n = 3).**

| NO. | Rt | Compound | Detection | | | |
|---|---|---|---|---|---|---|
| | | | Cultured | *Eisenia* | Barren | Sato et al. [18] |
| | | **Alcohols** | | | | |
| 8 | 12.05 | Methanol | + | + | + | + |
| 11 | 13.21 | 2-propanol | + | + | + | + |
| 12 | 13.46 | Ethanol | + | + | + | + |
| 20 | 16.57 | 2-butanol | + | + | + | + |
| 22 | 17.04 | Propanol | + | + | + | |
| 31 | 21.00 | Butanol | + | + | + | |
| 45 | 30.41 | 2-ethylhexanol | + | + | + | + |
| | | **Aldehydes** | | | | |
| 1 | 5.87 | Acetaldehyde | + | + | + | + |
| 2 | 6.74 | Dimethoxymethane | + | + | + | + |
| 7 | 11.78 | Acetal | + | + | + | + |
| 33 | 21.97 | 2-ethylhexanal | + | + | + | |
| 39 | 25.10 | Octanal | + | + | + | |
| 42 | 28.12 | Nonanal | + | + | + | + |
| 46 | 30.92 | Decanal | + | + | + | + |
| 47 | 32.01 | Benzaldehyde | + | + | + | + |
| | | **Aromatic compounds** | | | | |
| 4 | 7.79 | Methylcyclohexane | + | + | + | + |
| 13 | 13.63 | Benzene | + | + | + | |
| 23 | 17.26 | Toluene | + | + | + | + |
| 27 | 19.87 | Ethylbenzene | + | + | + | + |
| 28 | 20.34 | Xylene | + | + | + | + |
| 35 | 23.13 | Ethyltoluene | + | + | + | |
| 36 | 23.77 | Trimethylbenzene | + | + | + | + |
| 37 | 24.23 | Styrene | + | + | + | + |
| 44 | 29.74 | Dichlorobenzene | + | + | + | + |
| | | **Esters** | | | | |
| 6 | 11.56 | Ethyl acetate | + | + | + | + |
| 14 | 14.80 | Propyl acetate | + | + | + | + |
| 24 | 17.48 | S-methyl thioacetate | + | + | | + |
| 25 | 18.18 | Butyl acetate | + | + | + | |
| | | **Halomethanes** | | | | |
| 10 | 12.92 | Dichloromethane | + | + | + | + |
| 15 | 15.39 | Trichloroethane | + | + | + | |
| 19 | 16.26 | Chloroform | + | + | + | + |
| 43 | 29.63 | Bromoform | + | + | | |
| | | **Hydrocarbons** | | | | |
| 16 | 15.53 | Decane | + | + | + | + |
| 21 | 16.68 | α-pinene | + | + | + | + |
| 26 | 18.56 | Undecane | + | + | + | + |
| 29 | 20.43 | Δ3-carene | + | + | + | |
| 32 | 21.28 | 2,5,6-trimethyloctane | + | + | + | |
| 34 | 22.15 | Limonene | + | + | + | + |
| 38 | 24.61 | Tridecane | + | + | + | + |
| | | **Ketones** | | | | |

*(Continued)*

**Table 1.** (Continued)

| NO. | Rt | Compound | Detection | | | |
| --- | --- | --- | --- | --- | --- | --- |
| | | | Cultured | *Eisenia* | Barren | Sato et al. [18] |
| 5 | 9.12 | Acetone | + | + | + | + |
| 9 | 12.13 | 2-butanone | + | + | + | + |
| 18 | 16.06 | 4-methyl-2-pentanone | + | + | + | + |
| 30 | 21.00 | 3-heptanone | + | + | + | |
| 41 | 26.59 | 6-methyl-5-hepten-2-one | + | + | + | |
| 48 | 35.20 | Acetophenone | + | + | + | + |
| | | **Others** | | | | |
| 3 | 7.09 | Dimethyl sulfide | + | + | + | + |
| 17 | 15.72 | Acetonitrile | + | + | + | + |
| 40 | 25.42 | Bis-(methylthio)-methane | + | + | + | + |

An explanation of the terms Cultured, *Eisenia* and Barren is provided with Fig 1.

Rt indicates retention time (min).

tingling in nose and fishy odor. Butyl acetate from the gonads of CSU and ESU were described as having a green aroma and putrid odor, respectively.

## Halomethanes

Four halomethanes, dichloromethane, trichloroethane, chloroform and bromoform, were identified. Bromoform was not detected in the gonads of BSU. Chloroform and bromoform from the gonads of CSU were described as having a green aroma. The peak area of bromoform from CSU was significantly larger than that in the other groups ($P < 0.05$).

## Hydrocarbons

Seven hydrocarbons, decane, α-pinene, undecane, Δ3-carene, 2,5,6-trimethyloctane, limonene and tridecane, were detected from each group. The odors of decane and undecane were described. There were no significant differences in hydrocarbon peak areas among each group. Decane and undecane from the gonads of ESU were described as having a fishy odor and oxidized seaweed-like and hey smells, respectively.

## Ketones

Six ketones, acetone, 2-butanone, 4-methyl-2-pentanone, 3-heptanone, 6-methyl-5-hepten-2-one and acetophenone, were detected from the gonads of each group. The odors of three ketones were described. There were no significant differences in ketone peak areas among each group. 2-Butanone from the gonads of CSU was described as having a peach-like aroma. 6-Methyl-5-hepten-2-one from the gonads of ESU was described as having citrus-like and sweet aromas.

## Other compounds

Three other compounds, dimethyl sulfide (DMS), acetonitrile and bis-(methylthio)-methane, were detected, and their odors were described. DMS from the gonads of BSU was described as having a putrid odor. The peak area of DMS from the gonads of BSU was markedly large compared to that from the other groups. Bis-(methylthio)-methane from the gonads of CSU was described as having a fishy odor.

Table 2. The peak areas × 10$^{-5}$ (mean ± S.E.) and descriptions of odor-active volatile organic compounds in *Mesocentrotus nudus* gonads of each group (n = 3).

| NO. | Rt | Compound name | Peak area (×10$^{-5}$) | | | | Description | | |
|---|---|---|---|---|---|---|---|---|---|
| | | | Cultured | *Eisenia* | Barren | p | Cultured | *Eisenia* | Barren |
| | | **Alcohols** | | | | | | | |
| 11 | 13.21 | 2-propanol | 6.94 ± 1.09 | 8.25 ± 1.27 | 10.29 ± 0.61 | 0.076 | Kelp, seaweed, hey smell | | |
| 12 | 13.46 | Ethanol | 456.02 ± 212.64 | 559.31 ± 221.89 | 503.28 ± 187.53 | 0.922 | | Mint | |
| 20 | 16.57 | 2-butanol | 1.72 ± 0.39 | 1.70 ± 0.41 | 2.80 ± 0.52 | 0.216 | Seafood, hey smell | Unpleasant, sea urchin | |
| 45 | 30.41 | 2-ethylhexanol | 616.30 ± 284.38 | 526.13 ± 52.00 | 566.66 ± 102.69 | 0.938 | Sea urchin, seafood | Sea urchin | |
| | | **Aldehydes** | | | | | | | |
| 1 | 5.87 | Acetaldehyde | 6.12 ± 1.93 | 3.95 ± 1.24 | 3.72 ± 0.75 | 0.392 | | Putrid | |
| 2 | 6.74 | Dimethoxymethane | 3.12 ± 0.54 | 2.65 ± 0.35 | 2.65 ± 0.65 | 0.755 | Green, acrid | Putrid | |
| 7 | 11.78 | Acetal | 1.77 ± 1.33 | 2.31 ± 2.57 | 0.93 ± 0.32 | 0.714 | Sweet | | |
| 33 | 21.97 | 2-ethylhexanal | 0.49 ± 0.09 | 0.21 ± 0.18 | 0.36 ± 0.01 | 0.119 | Sweet | | |
| 39 | 25.10 | Octanal | 0.15 ± 0.01 | 0.14 ± 0.04 | 0.19 ± 0.02 | 0.193 | Green | Putrid | Green |
| 42 | 28.12 | Nonanal | 2.31 ± 0.16 | 2.54 ± 0.54 | 2.60 ± 0.29 | 0.723 | Plastic, medicinal | | |
| 46 | 30.92 | Decanal | 2.74 ± 0.39 | 2.51 ± 0.51 | 2.45 ± 0.29 | 0.813 | | | Green |
| 47 | 32.01 | Benzaldehyde | 0.22 ± 0.06 | 0.21 ± 0.05 | 0.23 ± 0.02 | 0.962 | Sea urchin | | |
| | | **Aromatic compounds** | | | | | | | |
| 4 | 7.79 | Methylcyclohexane | 3.29 ± 0.27 | 2.74 ± 0.89 | 4.05 ± 0.39 | 0.150 | Benzene | | |
| 13 | 13.63 | Benzene | 0.62 ± 0.05 | 0.62 ± 0.17 | 0.66 ± 0.15 | 0.958 | Tingle in nose, heave note | Mint | |
| 23 | 17.26 | Toluene | 71.64 ± 16.58 | 54.25 ± 17.59 | 59.44 ± 7.81 | 0.606 | Fruit | Sweet | |
| 27 | 19.87 | Ethylbenzene | 54.44 ± 10.42 | 61.95 ± 9.12 | 58.98 ± 10.80 | 0.873 | Sea urchin, baked fish | Putrid fish, sour | |
| 36 | 23.77 | Trimethylbenzene | 1.03 ± 0.09 | 1.26 ± 0.41 | 1.07 ± 0.06 | 0.542 | Hey smell | | |
| 37 | 24.23 | Styrene | 0.68 ± 0.12[b] | 0.86 ± 0.16[b] | 1.91 ± 0.08[a] | < 0.001 | Hey smell | Hey smell | |
| 44 | 29.74 | Dichlorobenzene | 4.43 ± 1.32 | 3.83 ± 1.50 | 6.62 ± 0.93 | 0.227 | Green | | |
| | | **Esters** | | | | | | | |
| 6 | 11.56 | Ethyl acetate | 11.76 ± 1.40 | 10.51 ± 1.28 | 8.65 ± 0.75 | 0.171 | Sweet | | |
| 14 | 14.80 | Propyl acetate | 3.00 ± 0.25 | 2.74 ± 1.47 | 2.33 ± 0.24 | 0.616 | Sweet, sweet candy | | |
| 24 | 17.48 | S-methyl thioacetate | 10.75 ± 9.61[a] | 0.38 ± 0.33[a] | ND[b] | < 0.001 | Tingle in nose, heavy note, fishy | | |
| 25 | 18.18 | Butyl acetate | 0.73 ± 0.13 | 0.62 ± 0.11 | 0.48 ± 0.12 | 0.327 | Green | Putrid | |
| | | **Halomethanes** | | | | | | | |
| 19 | 16.26 | Chloroform | 8.17 ± 1.82 | 5.61 ± 1.01 | 6.55 ± 1.94 | 0.542 | Green, hey smell | | Plastic, medicinal |
| 43 | 29.63 | Bromoform | 0.12 ± 0.04[a] | 0.04 ± 0.01[b] | ND[c] | < 0.001 | Green | | |
| | | **Hydrocarbons** | | | | | | | |
| 16 | 15.53 | Decane | 2.02 ± 0.41 | 2.56 ± 1.47 | 2.32 ± 0.53 | 0.833 | | Fishy | |
| 26 | 18.56 | Undecane | 2.25 ± 0.17 | 2.10 ± 0.57 | 2.67 ± 0.50 | 0.545 | Tingle in nose, heavy note | Oxidized seaweed, hey smell | |
| | | **Ketones** | | | | | | | |
| 5 | 9.12 | Acetone | 116.18 ± 45.05 | 103.60 ± 20.71 | 113.94 ± 9.31 | 0.926 | | | Sweet |
| 9 | 12.13 | 2-butanone | 11.62 ± 2.92 | 9.41 ± 3.49 | 7.85 ± 1.45 | 0.521 | Peach | | |
| 41 | 26.59 | 6-methyl-5-hepten-2-one | 2.61 ± 0.72 | 1.87 ± 1.13 | 2.29 ± 0.70 | 0.761 | | Sweet, citrus fruit | |
| | | **Others** | | | | | | | |
| 3 | 7.09 | Dimethyl sulfide | 16.18 ± 7.39 | 9.47 ± 1.59 | 65.17 ± 48.88 | 0.261 | | | Putrid |
| 17 | 15.72 | Acetonitrile | 48.98 ± 20.79 | 44.59 ± 18.81 | 56.33 ± 17.72 | 0.887 | | | Tree bark |

(*Continued*)

**Table 2.** (Continued)

| NO. | Rt | Compound name | Peak area (×10⁻⁵) | | | *p* | Description | | |
|---|---|---|---|---|---|---|---|---|---|
| | | | Cultured | *Eisenia* | Barren | | Cultured | *Eisenia* | Barren |
| 40 | 25.42 | Bis-(methylthio)-methane | 2.32 ± 0.96 | 1.92 ± 0.92 | 0.97 ± 0.11 | 0.257 | Fishy | | |

Significance of values in peak areas among groups analyzed by ANOVA are provided. ND indicate no detection. An explanation of the terms Cultured, *Eisenia* and Barren is provided in Fig 1. Lower-case letters indicate significant differences in peak areas among groups ($p < 0.05$, Tukey's test). An explanation of the term Rt is provided with Table 1.

## Discussion

In the present study, 2-butanol, 2-ethylhexanol, benzaldehyde and ethylbenzene detected from the gonads of *M. nudus* were described as having sea urchin-like aromas for the first time. Benzaldehyde has pleasant almond, nutty and stony fruit like aromas of peach (*Prunus persica*) [26]. De Quirós et al. [17] suggested that this compound is associated with the pleasant aroma of sea urchin. Ethylbenzene is detected in the gonads of some sea urchin species [17, 18]. Phillips et al. [16] suggested a positive correlation between ethylbenzene concentration and marine odor. Sato et al. [18] reported that ethylbenzene from the gonads of *M. nudus* in Naburi Bay was described as having a fish oil scent, which would lead to the pleasant aroma of fresh sea urchin gonads. In the present study, ethylbenzene from the gonads of CSU was described as having sea urchin-like and baked fish aromas, and that from ESU was described as having putrid fish-like and sour odors with slightly larger peak areas than those of CSU.

A larger number of sweet and fruity pleasant odor descriptions (acetal, 2-ethylhexanal, toluene, ethyl acetate, propyl acetate, acetone, 2-butanone and 6-methyl-5-hepten-2-one described) detected from the gonads of cultured sea urchins than from those of wild sea urchins would reflect the sweet taste of their gonads [8]. Esters are associated with the sweet aroma of various fruits [27]. Propyl acetate detected from gonads of *M. nudus* in Naburi Bay was described as having a sweet aroma [18]. Toluene can be detected from some fruits: strawberry, apple, grape, mango and sapodilla [28]. Toluene from sapodilla fruit (*Achras sapota*) and mango (*Mangifera indica*) are described as having sweet and caramel aromas, and caramel and solvent odors, respectively [29, 30]. Ketones from crustaceans have sweet floral and fruity flavors [31]. 2-Butanone can be detected from pineapple [32]. This compound is also detected from yellow passion fruit and has fruity, moldy, woody, fresh and bitter aromas [33]. These past studies suggest that propyl acetate, toluene and 2-butanone influence the fresh and fruity sweet aromas of the gonads of CSU.

The odor descriptions of dimethoxymethane, octanal, benzene, ethylbenzene, butyl acetate, and undecane differed between gonads from CSU and ESU. Of them, octanal is contained in several seaweeds [34–36] and the octanal content differs among seaweed species [35]. Sato et al. [18] suggested that octanal in *M. nudus* gonads would be derived from their consumption of seaweeds as food. Differences in seaweed species consumed between CSU and ESU changed the odor description of octanal. The differences in odor descriptions of other compounds remain to be identified.

Sulfur-containing compounds affect the overall flavor because of their low thresholds [37]. Bis-(methylthio)-methane has a garlic-like odor, which is one of the off-flavor components of prawns and sand lobsters [38]. This compound is produced by a metabolite of *Shewanella putrefaciens* grown on high-pH pork [39]. This compound from the gonads of *M. nudus* in Naburi Bay was described as having ozone and sulfur odors [18] in comparison to having a fishy odor when extracted from the gonads of CSU in the present study. These results suggest

that bis-(methylthio)-methane can be one of the factors affecting the unpleasant odor of decomposed shellfish. DMS is known to have a sulfur odor when extracted from mussels [40] and oysters [41]. This compound from the gonads of *M. nudus* in Naburi Bay was described as having marine and fish-like odors [18]. From the gonads of *E. chloroticus*, DMS is also associated with marine, seafood and sharp odors [16]. In the present study, this compound from the gonads of BSU was described as having a strong putrid odor, and the peak area was markedly large regardless of a lack of an odor description from the gonads of other groups. This compound is detected from several fish and shellfish species [42, 43]. Therefore, *M. nudus* from the barren would reflect the omnivorous food habit [20] because there is no erect algae on barrens, and the sea urchins would consume dead fish or other shellfish [18].

In conclusion, from the gonads of cultured *M. nudus*, a larger number of odor-active VOCs with a green scent and sweet aromas were detected than from those of wild sea urchins, which would enhance the pleasant aroma and richness of flavor, leading to high sensory evaluation [8]. The present study demonstrated the effect of *Saccharina* kelp feeding on the odor-active compound profile of gonads of edible sea urchin for the first time. The number of VOCs detected from the gonads of cultured and wild *M. nudus* was almost the same. However, kelp feeding decreased the strong, putrid odor from DMS, enhanced pleasant, sweet aroma, and would enhance the richness and sweetness of flavor. The appropriate feeding duration for improvement in gonad flavors must be further studied.

## Supporting information

**S1 Table. Raw data of chromatogram and peak area of volatile organic compounds from gonads of *Mesocentrotus nudus* of each group.**
(XLSX)

## Acknowledgments

We are grateful to the Operative Chief, N. Sasaki, the Head of the Youth Division, Y. Sugawara, and other staff of the Shizugawa Branch of the Miyagi Fisheries Cooperative Association for their full cooperation with managing the sea urchin cage cultures. We sincerely thank S. Owada of the Hirota Fisheries Cooperative Association for providing *S. japonica*. We also thank M. Oshima and S. Kodama of Diving Stage Ariel for collecting sea urchins for culturing.

## Author Contributions

**Data curation:** Satomi Takagi, Yoichi Sato, Atsuko Kokubun, Eri Inomata.

**Formal analysis:** Satomi Takagi, Yoichi Sato, Atsuko Kokubun.

**Funding acquisition:** Satomi Takagi, Yoichi Sato, Atsuko Kokubun, Yukio Agatsuma.

**Resources:** Satomi Takagi, Eri Inomata, Yukio Agatsuma.

**Supervision:** Yukio Agatsuma.

**Validation:** Yukio Agatsuma.

**Writing – original draft:** Satomi Takagi.

**Writing – review & editing:** Yoichi Sato, Atsuko Kokubun, Eri Inomata, Yukio Agatsuma.

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
