## [Decision Letter · Decision Letter 0]

24 Feb 2020

PONE-D-20-02508

Odor-active compounds from the gonads of Mesocentrotus nudus sea urchins fed Saccharina japonica kelp

PLOS ONE

Dear Yukio Agatsuma,

Thank you for submitting your manuscript to PLOS ONE. After careful consideration, we feel that it has merit but does not fully meet PLOS ONE’s publication criteria as it currently stands. Therefore, we invite you to submit a revised version of the manuscript that addresses the points raised during the review process.

ACADEMIC EDITOR: 

Please verify carefully all the comments of the reviewers expecially on the "number of pannelist".

We would appreciate receiving your revised manuscript by 25 March 2020. To enhance the reproducibility of your results, we recommend that if applicable you deposit your laboratory protocols in protocols.io, where a protocol can be assigned its own identifier (DOI) such that it can be cited independently in the future. For instructions see: http://journals.plos.org/plosone/s/submission-guidelines#loc-laboratory-protocols

We look forward to receiving your revised manuscript.

Kind regards,

Filippo Giarratana

Academic Editor

PLOS ONE

Journal Requirements:

"A part of this work was financially supported by the Grand-in-Aid for JSPS Fellow [grant number 17J02308] from Japan Society for the Promotion of Science to S.T."

We note that one or more of the authors are employed by a commercial company: RIKEN VITAMIN Co., Ltd. and Riken Food Co., Ltd

Reviewers' comments:

Reviewer's Responses to Questions

**Comments to the Author**

1. Is the manuscript technically sound, and do the data support the conclusions?

Reviewer #1: Partly

Reviewer #2: Partly

2. Has the statistical analysis been performed appropriately and rigorously? 

Reviewer #1: Yes

Reviewer #2: I Don't Know

3. Have the authors made all data underlying the findings in their manuscript fully available?

Reviewer #1: Yes

Reviewer #2: Yes

4. Is the manuscript presented in an intelligible fashion and written in standard English?

Reviewer #1: Yes

Reviewer #2: Yes

5. Review Comments to the Author

Reviewer #1: This is an interesting study. It addresses a little studied aspect of the attractiveness of sea urchin gonads, i.e. its aroma. It asks whether aroma is caused by aromatic compounds in the gonads and that these are affected by the food the sea urchin eats. The documentation that the aromatic compounds of gonads of sea urchins from different environments and individuals fed kelp differ is excellent. This is of interest in itself. It has implications for intermediary metabolism of sea urchins.

.However I have some questions.

My major concern is the analysis of the aroma of the gonads. The authors state “141 GC-sniffing analysis The 146 panelist recorded the retention time and the related description of the aroma compounds.

How many panelists were there? The statement implies there was only one. If so, this is inadequate, and the paper should be rejected. One person is not a panel.

Other concerns and questions are less important.

1. 90 prefecture (38°40´N, 141°30´E) on 6 May 2016

What was the gonadal stage? Did it differ between the groups? Could gonadal stage affect odor?

2. 107 gonads were randomly collected from one group of gonads and stored in a polystyrene

108 storage container at 4 °C until further analyses.

How long before “further analyses”?

3. 108 Three containers were prepared for each group (n = 3).

Table 1. Volatile organic compounds (VOCs) detected 181 from gonads of Mesocentrotus

182 nudus using gas chromatography-mass spectrometry (n = 3). The table does not indicate variability. The results are given only as + or –. Were the results the same for all three samples?

4. The descriptions are qualitative.

Some descriptions are not understandable: green, gas, thinner, heavy and other descriptive terms must be given.

Sweet, sweet candy is not descriptive of aroma. Sweet is a qualitative indicator of taste, not aroma.

Unpleasant, putrid, sour, fishy are good descriptions of aroma.

5. The authors use both “aroma” and “odor”. They should be consistent. The words are synonyms, although the dictionary states “odor” is considered to indicate unpleasant in the US.

In this regard, perhaps this statement in the abstract should be changed:

Kelp feeding decreased the strong, putrid odor from 34 dimethyl sulfide, increased the number of odor-active compounds, particularly those with 35 a sweet aroma, and enhanced the richness and sweetness of flavor

The abstract should concern only the results found in the study. The reference to “enhanced “enhanced the richness and sweetness of flavor” is from other studies and should be in the Discussion where it is pertinent.

Reviewer #2: The MS is on the whole clear and easy to read, however there is some essential information missing as well as information and discussion that would greatly enhance the paper.

1) Details on the GCMS method are missing. How were the RI of the volatiles calculated ? what standards were used ? Presumably these were then matched to the RI’s of the compounds obtained by the GC-O

2) How were the panellists trained ? How many panellists were used ? How many replicate sniffs ? The descriptors they used seem to be very limited. Odour descriptors such as “sweet, Gas, thinner, heavy, dry are not very informative. What is meant by these? Also “oxidised seaweed” what the standard odour used to train panellist on this ? In fact all odours used for panel training should be described. Panel training is best practice for GC-O

3) How was retention time assessed ? i.e. what actions did the panellist take – key pad ? finger span?

4) What were the size of the urchins ? roe weight ? yield – presumably the diet affected these values ?

5) What was the gender of the urchins ? is this a male / female difference rather than a diet difference being reported

6) What was the appearance – shape, colour of the roe obtained from the urchins on the three diets.

7) L 280. I find the comparison between sea urchin and Benzaldehyde as almond, nutty and stone fruit to be perplexing – are these typical descriptors for urchin roe.

8) L287 – Describing the same compound from two different sources as too different odours, could be due to a concentration affect, but I am also concerned about compound identification and panellist training – please include an explanation for this observation in the text.

9) L294 – “reflect the sweet taste of their gonads??? Were the gonads tasted.

10) Table 2 define ND – also does ND mean that it was not present ? what was the detection limit. Why it <0.001 used in the Sato et al data – presumably this also means that it is ND.

11) I found the discussion in general to be a little limited, particularly with reference and published literature.

12) Why / how do the diets impact on aroma – what is in the diets – what is the mechanism that generates different compounds – do compounds differ or do the differences in odour reflect differences in concentration – is so what are trends in VOC.

13) Did sampling only occur at a single timepoint ? Sampling over time would have provided useful information

14) L 330 – larger number of VOC with unpleasant odours …. Were detected …. which enhanced the pleasant aroma and richness of flavour, leading to high sensory evaluation. First of all what would more unpleasant odours improve taste ? How was taste assessed ? If this MS is part of a larger study – this neds to be made clearer in the MS and some data produced to support the statements.

15) The MS seemed to lack a conclusion

6. PLOS authors have the option to publish the peer review history of their article (what does this mean?). If published, this will include your full peer review and any attached files.

Reviewer #1: No

Reviewer #2: No

---

## [Author Response · Author response to Decision Letter 0]

27 Mar 2020

Response to Reviewers

Reviewer #1: This is an interesting study. It addresses a little studied aspect of the attractiveness of sea urchin gonads, i.e. its aroma. It asks whether aroma is caused by aromatic compounds in the gonads and that these are affected by the food the sea urchin eats. The documentation that the aromatic compounds of gonads of sea urchins from different environments and individuals fed kelp differ is excellent. This is of interest in itself. It has implications for intermediary metabolism of sea urchins.

・We appreciate your comments.

However I have some questions.

My major concern is the analysis of the aroma of the gonads. The authors state “141 GC-sniffing analysis The 146 panelist recorded the retention time and the related description of the aroma compounds.

How many panelists were there? The statement implies there was only one. If so, this is inadequate, and the paper should be rejected. One person is not a panel.

・Certainly. We had three panelists. The “panelist” was not correct. We changed the sentence “The panelist recorded the retention time and the related description of the aroma compounds” to “The three panelists, who are well versed in sea urchin gonad quality and share common perceptions, recorded the retention time and the related description of the aroma compounds by writing in a paper” on the lines 152–156 in Materials and methods.

We apologize for the mistake.

Other concerns and questions are less important.

1. 90 prefecture (38°40´N, 141°30´E) on 6 May 2016

What was the gonadal stage? Did it differ between the groups? Could gonadal stage affect odor?

・Takagi et al. (2019) showed the gonad developmental stages of the sea urchins which were collected and cultured in the same manner with sea urchins used in the present study. Almost gonads of the sea urchins from all groups were in the growing stage. Therefore, we think there is no need to consider the effects of gonad developmental stages. We added the sentence “Takagi et al. [8] showed no significant differences in test diameter among these sea urchins (CSU, ESU and BSU), and almost the gonads were in the growing stage” on the lines 113–115 in Materials and Methods.

2. 107 gonads were randomly collected from one group of gonads and stored in a polystyrene

108 storage container at 4 °C until further analyses.

How long before “further analyses”?

・Certainly, we should mention about that. Thank you for the comment. GC-MS and GC-sniffing analyses were conducted within 48 hours after dissection. We added “within 48 hours after dissection” at the end of the sentence “Headspace sampling, GC-MS analysis and GC-sniffing analysis were conducted according to Sato et al. [18]” on the lines 112–113 in Materials and methods.

3. 108 Three containers were prepared for each group (n = 3).

Table 1. Volatile organic compounds (VOCs) detected 181 from gonads of Mesocentrotus

182 nudus using gas chromatography-mass spectrometry (n = 3). The table does not indicate variability. The results are given only as + or –. Were the results the same for all three samples?

・As you mentioned, we prepared three containers for each group and conducted analyses per container. When a VOC was detected in triplicate analysis, this determined the presence of the VOC in the group as we mentioned on the lines 142–143 in Materials and methods. 

4. The descriptions are qualitative.

Some descriptions are not understandable: green, gas, thinner, heavy and other descriptive terms must be given.

・Thank you for your comments. Green was used for the description of odor of sea urchin gonads in a past study (Niimi et al. 2010). We changed “gas” to “plastic, medicinal”, “thinner” to “benzene”, “heavy” to “tingle in nose, heavy note” in Table 2. In addition, we changed the sentences “Benzene from the gonads of CSU was described as having a heavy odor, although that from ESU was described as having a mint aroma” to “Benzene from the gonads of CSU was described as having a heavy note, although that from ESU was described as having a mint aroma” on the lines 224–225, “S-methyl thioacetate from the gonads of CSU showed a markedly larger peak area than those of other groups and was described as having a heavy and fishy odor” to “S-methyl thioacetate from the gonads of CSU showed a markedly larger peak area than those of other groups and was described as having a heavy note tingling in nose and fishy odor” on the lines 239–241 in Results.

Sweet, sweet candy is not descriptive of aroma. Sweet is a qualitative indicator of taste, not aroma.

・Sweet was also used as a description of aroma in past studies (e.g., Niimi et al. 2010; Lehtinen & Veijanen 2011). 

Unpleasant, putrid, sour, fishy are good descriptions of aroma.

・We appreciate.

5. The authors use both “aroma” and “odor”. They should be consistent. The words are synonyms, although the dictionary states “odor” is considered to indicate unpleasant in the US.

In this regard, perhaps this statement in the abstract should be changed:

Kelp feeding decreased the strong, putrid odor from 34 dimethyl sulfide, increased the number of odor-active compounds, particularly those with 35 a sweet aroma, and enhanced the richness and sweetness of flavor

The abstract should concern only the results found in the study. The reference to “enhanced “enhanced the richness and sweetness of flavor” is from other studies and should be in the Discussion where it is pertinent.

・We appreciate your comments. As you mentioned we changed the sentence “Kelp feeding decreased the strong, putrid odor from dimethyl sulfide, increased the number of odor-active compounds, particularly those with a sweet aroma, and enhanced the richness and sweetness of flavor” to “Kelp feeding decreased the putrid odor from dimethyl sulfide, and enhanced pleasant, sweet aromas” on the line 33–34 in the abstract, and changed the sentence “However, kelp feeding decreased the strong, putrid odor from DMS, increased the number of odor-active compounds, particularly those with a sweet aroma, and enhanced the richness and sweetness of flavor” to “However, kelp feeding decreased the strong, putrid odor from DMS enhanced pleasant, sweet aroma, and would enhance the richness and sweetness of flavor” on the lines 332–333 in Discussion.

Thank you very much.

Reviewer #2: The MS is on the whole clear and easy to read, however there is some essential information missing as well as information and discussion that would greatly enhance the paper.

1) Details on the GCMS method are missing. How were the RI of the volatiles calculated? what standards were used? Presumably these were then matched to the RI’s of the compounds obtained by the GC-O

・There are past studies of GC-MS analysis published without RI (e.g., De Quirós et al. 2001; Zhu et al. 2009; de Alencar et al. 2017). We identified each VOC from retention time by similarity search (Zhu et al. 2009) using the library of Powered Pro software (Wiley 11N17main, Agilent Technologies Inc.). We added the sentence “Each VOC was identified by similarity search [25] using the library of the software (Wiley 11N17main, Agilent Technologies Inc.)” on the lines 141–142 in Materials and methods, and added Zhu et al. (2009) to the reference list. According to the reference addition, the reference NO. of other references was changed.

2) How were the panellists trained? How many panellists were use? How many replicate sniffs? The descriptors they used seem to be very limited. Odour descriptors such as “sweet, Gas, thinner, heavy, dry are not very informative. What is meant by these? Also “oxidised seaweed” what the standard odour used to train panellist on this? In fact all odours used for panel training should be described. Panel training is best practice for GC-O

・Thank you for your advices. One of the purposes of this study is to identify the compounds with sea urchin like aroma. Therefore, we prepared three panelists who are well versed in sea urchin gonad quality and share common perceptions. We added the sentence “The purpose of this study is to (1) clarify the effect of S. japonica feeding to improve gonad flavor of sea urchins from a barren and (2) identify the compounds with sea urchin like aroma” on the lines 78–80 in Introduction, and changed the sentence “The panelist recorded the retention time and the related description of the aroma compounds” to “The three panelists, who are well versed in sea urchin gonad quality and share common perceptions, recorded the retention time and the related description of the aroma compounds by writing in a paper” on the lines 152–155 in Materials and Methods. 

“Sweet” is used as a description of odor in past studies (e.g., Phillips et al. 2010; Niimi et al. 2010; Lehtinen & Veijanen 2011). We changed “gas” to “plastic, medicinal”, “thinner” to “benzene”, “heavy” to “tingle in nose, heavy note”, and “dry” to “hey smell” in Table 2. In addition, we changed the sentences “Benzene from the gonads of CSU was described as having a heavy odor, although that from ESU was described as having a mint aroma” to “Benzene from the gonads of CSU was described as having a heavy note, although that from ESU was described as having a mint aroma” on the lines 224–225, “S-methyl thioacetate from the gonads of CSU showed a markedly larger peak area than those of other groups and was described as having a heavy and fishy odor” to “S-methyl thioacetate from the gonads of CSU showed a markedly larger peak area than those of other groups and was described as having a heavy note tingling in nose and fishy odor” on the lines 239–241, “The odors of 2-butanol from the gonads of CSU and ESU were described as acceptable seafood and dry odors and as an unpleasant sea urchin-like odor, respectively” to “The odors of 2-butanol from the gonads of CSU and ESU were described as acceptable seafood and hey smells and as an unpleasant sea urchin-like odor, respectively” on the lines 203–204, “2-Propanol from the gonads of CSU was described as having kelp and dry odors” to “2-Propanol from the gonads of CSU was described as having kelp and hey smells” on the lines 206–207, " Styrene from the gonads of CSU and ESU was described as having a dry odor” to “Styrene from the gonads of CSU and ESU was described as having a hey smell”, on the lines 231–232, and “Decane and undecane from the gonads of ESU were described as having a fishy odor and oxidized seaweed-like and dry odors, respectively” to “Decane and undecane from the gonads of ESU were described as having a fishy odor and oxidized seaweed-like and hey smells, respectively” on the lines 256–257 in Results. Thank you again.

3) How was retention time assessed? i.e. what actions did the panellist take – key pad? finger span?

・The panelists could watch the screen which shows elapsed time during the sniffing analysis. They recorded the retention time by writing in a paper. They could conduct the record without taking off their nose from the sniffing port. We added the sentence “They could record without taking off their nose from the sniffing port” on the lines 155–156 in Materials and methods, and changed the sentence “The panelist recorded the retention time and the related description of the aroma compounds” to “The three panelists, who are well versed in sea urchin gonad quality and share common perceptions, recorded the retention time and the related description of the aroma compounds by writing in a paper” on the lines 152–155 in Materials and Methods.

4) What were the size of the urchins? roe weight? yield – presumably the diet affected these values?

・The sea urchins on barrens have small and low quality gonads, showing low commercial value. For that reason, the gonads size per urchin varies among the three groups. One of the purposes of the culture experiment is to clarify the effect of S. japonica feeding on improvement in gonad flavor of sea urchins from barrens. In this study, we used the sea urchins with same size (ca. 50 mm test diameter) for each treatment, and ca. 5 g gonads were collected randomly from 20 individuals of each group and repeated three times.

We added the sentences “Takagi et al. [8] conducted a sensory evaluation and measurements and analyses of gonad size, developmental stages, color, texture and free amino acid contents of sea urchins cultured and collected from the same algal beds by the same methods on the same day with the present study. The purpose of this study is to (1) clarify the effect of S. japonica feeding to improve gonad flavor of sea urchins from a barren and (2) identify the compounds with sea urchin like aroma” on the lines 75–80 in Introduction, and “Takagi et al. [8] showed no significant differences in test diameter among these sea urchins (CSU, ESU and BSU), and almost the gonads were in the growing stage” on the lines 113–115 in Materials and methods. We also added “therefore no commercial value [20]” at the end of the sentence “This species densely distributes on barrens [20], where the gonads show small size [21] and deteriorated color, texture, and taste [13, 22]” on the lines 59–60 in Introduction.

5) What was the gender of the urchins? is this a male / female difference rather than a diet difference being reported

・In the present study, the gonads of sea urchins were in the growing stage (Takagi et al. 2019), but were not observed by sex because of no histological observation in the present study Because we had to conduct GC-MS and GC-sniffing analyses immediately after dissection to analyze fresh gonads. Furthermore, because gonads of sea urchins from the barren were too small to conduct the analyses individually, some individuals of each group have to be combined for the analyses (therefore we collected 5 g of gonads from 20 individuals of each group with three replication). Thus, it was impossible to separate sex in the present study. 

6) What was the appearance – shape, colour of the roe obtained from the urchins on the three diets.

・Takagi et al. (2019) reported gonad size, color, texture and taste of the sea urchins. We changed the sentence “Takagi et al. [8] first improved the gonad taste of barren individuals to a more desirable level compared to that of individuals from an Eisenia kelp bed (fishing ground) by feeding them fresh cultivated Saccharina japonica during May–July” to “Takagi et al. [8] first improved the whole gonad quality (size, color, texture and taste) of barren individuals to a more desirable level compared to that of individuals from an Eisenia kelp bed (fishing ground) by feeding them fresh cultivated Saccharina japonica during May–July” on the lines 63–66, and added the sentence “Takagi et al. [8] conducted a sensory evaluation and measurements and analyses of gonad size, developmental stages, color, texture and free amino acid contents of sea urchins cultured and collected from the same algal beds by the same methods on the same days with the present study” on the lines 75–78 in Introduction.

7) L 280. I find the comparison between sea urchin and Benzaldehyde as almond, nutty and stone fruit to be perplexing – are these typical descriptors for urchin roe.

・Thank you for your comments. Since there are only four papers which conducted odor analyses of sea urchin gonads, it is difficult to judge whether the descriptions are typical for sea urchin gonads or not. “Almond” is also used as a description of sea urchin gonad odor in a past study which conducted a sensory evaluation (Phillips et al. 2009). Thus, we think using these descriptors are no problem.

8) L287 – Describing the same compound from two different sources as too different odours, could be due to a concentration affect, but I am also concerned about compound identification and panellist training – please include an explanation for this observation in the text.

・Referring to Question (2), the panelists of the sniffing analysis are well versed in sea urchin gonad quality and share common perceptions.

9) L294 – “reflect the sweet taste of their gonads??? Were the gonads tasted.

・As we mentioned on the lines 67–69 in Introduction, Takagi et al. (2019) evaluated the quality of the gonads of sea urchins collected in the same manner with CSU, ESU and BSU by a sensory test, and the gonads of CSU and ESU were evaluated sweeter than that of BSU. We suggested the sweet aromas could be affected by the strong sweetness of gonads of cultured sea urchins because flavor and odor could influence the taste of food.

10) Table 2 define ND – also does ND mean that it was not present? what was the detection limit. Why it <0.001 used in the Sato et al data – presumably this also means that it is ND.

・We apologize for making confusion. We did not refer to Sato et al. (2019) in Table 2. We provided the significant values in peak areas among groups analyzed by ANOVA in Table 2. We added the sentences “Significance of values in peak areas among groups analyzed by ANOVA are provided. ND indicate no detection” to the legend of Table 2. We apologize for no description in the previous MS. 

11) I found the discussion in general to be a little limited, particularly with reference and published literature.

・Certainly, but there are only four paper which conducted odor analyses of sea urchin gonads (De Quirós et al. 2001; Niimi et al. 2010; Phillips et al. 2009; Sato et al. 2019), resulting in a limitation of references. 

12) Why / how do the diets impact on aroma – what is in the diets – what is the mechanism that generates different compounds – do compounds differ or do the differences in odour reflect differences in concentration – is so what are trends in VOC.

・Referring to Question 11), there are only four studies analyzed odor of sea urchin gonads. Phillips et al. (2009) revealed that the odors of gonads of sea urchin collected from the North Island and the South Island of New Zealand varied but they did not mention about the reason. We used sea urchins cultured and fed fresh S. japonica kelp (CSU), and two types of wild sea urchins collected from a barren (BSU) and an Eisenia kelp bed (ESU). Past studies revealed that the difference in the habitat of sea urchins results in the variation of water temperature and the seaweed vegetation as sea urchin food, and these affect to the variation of body size, growth rate, and gonad development, color, texture and taste of sea urchins (e.g., Addis et al. 2015; Agatsuma et al. 2005; Ling et al. 2010; Takagi et al. 2017). In the present study, the culture and collection of sea urchins in the three groups were conducted in the same bay. Thus, the variation of gonad odor among the three groups could result from the food. 

As you mentioned, we deleted the sentence “These results suggest that an extremely high content of ethylbenzene could result in an unpleasant odor of gonads” on the lines 286–287 because we do not have the confidence of effect of peak areas on the description.

We appreciate your valuable comments.

13) Did sampling only occur at a single timepoint? Sampling over time would have provided useful information

・Thank you for the comment. As we answered for Question (4), one of the purposes of the present study is to clarify the effect of S. japonica feeding to improve gonad flavor of sea urchins from a barren. In addition, the fishing season of M. nudus is between June and August (Kawamura 1993). Thus, sea urchin collection was conducted on 19 July 2016, at the end of the culture experiment. 

14) L 330 – larger number of VOC with unpleasant odours …. Were detected …. which enhanced the pleasant aroma and richness of flavour, leading to high sensory evaluation. First of all what would more unpleasant odours improve taste? How was taste assessed? If this MS is part of a larger study – this neds to be made clearer in the MS and some data produced to support the statements.

・As we answered for Question (9), Takagi et al. (2019) conducted a sensory evaluation on the sea urchins. We added the sentence “Takagi et al. [8] conducted a sensory evaluation and measurements and analyses of gonad size, gonad color, texture and free amino acid contents of sea urchins cultured and collected from the same algal bed by the same methods on the same day with the present study” on the lines 75–78 in Introduction.

Following your suggestion, we changed the sentence “From the gonads of cultured M. nudus, a larger number of odor-active VOCs with unpleasant odors, a green scent, and sweet aromas were detected than from those of wild sea urchins, which enhanced the pleasant aroma and richness of flavor, leading to high sensory evaluation [8]” to “In conclusion, from the gonads of cultured M. nudus, a larger number of odor-active VOCs with a green scent and sweet aromas were detected than from those of wild sea urchins, which would enhance the pleasant aroma and richness of flavor, leading to high sensory evaluation [8]” on the lines 326–329 in Discussion.

Thank you very much.

15) The MS seemed to lack a conclusion

・Certainly. We changed the sentence “From the gonads of cultured M. nudus, a larger number of odor-active VOCs with unpleasant odors, a green scent, and sweet aromas were detected than from those of wild sea urchins, which enhanced the pleasant aroma and richness of flavor, leading to high sensory evaluation [8]” to “In conclusion, from the gonads of cultured M. nudus, a larger number of odor-active VOCs with a green scent and sweet aromas were detected than from those of wild sea urchins, which would enhance the pleasant aroma and richness of flavor, leading to high sensory evaluation [8]” on the lines 326–329 in Discussion.

Thank you for your valuable, insightful comments.

References

Addis P, Moccia D, Secci M (2015) Mar. Ecol. 36: 178–184

Agatsuma Y, Sato M, Taniguchi K (2005) Aquaculture 249: 449–458

de Alencar DB, Diniz JC, Rocha SAS, Pires-Cavalcante KMS, Freitas JO, Nagano CS, Sampaio AH, Saker-Sampaio S (2017) J. Appl. Phycol. 29: 1571–1576

De Quirós ARB, López-Hernández J, González-Castro MJ, de la Cruz-García C, Simal-Lozano J. (2001) Eur. Food Res. Technol. 212: 643–647

Kawamura (1993) Uni Zouyoushoku to Kakou Ryutsu. Sapporo, Japan: Hokkai Suisan Shinbun Company. 253 pp (in Japanese)

Lehtinen J, Veijanen A (2011) Water Air Soil Pollut. 218: 185–196

Ling SD, Ibbott S, Sanderson (2010) J. Exp. Mar. Biol. Ecol. 395: 135–146

Niimi J, Leus M, Silcock P, Hamid N, Bremer P. (2010) Food. Chem. 121: 601–607.

Phillips K, Bremer P, Silcock P, Hamid N, Delahunty C, Barker M et al. (2009) Aquaculture 288: 205–215. 

Phillips K, Niimi J, Hamid N, Silcock P, Delahunty C, Barker M, et al. (2010) LWT-Food Sci. Technol. 2010; 43: 202–213.

Sato Y, Takagi S, Inomata E, Agatsuma Y. (2019) Food and Nutrition Sciences 10: 860–875.

Takagi S, Murata Y, Inomata E, Aoki MN, Agatsuma Y (2019) J. Appl. Phycol. 31: 4037–4048.Zhu X, Gao Y, Chen Z, Su Q (2009) Chromatographia 69: 735–742.

Other changes

We changed the sentences “The data below the detection limits were inputted as 0.0001” to “The values of not detected compounds were inputted as 0.0001.” on the lines 162–163 in Materials and Methods.

We changed the sentence “Benzaldehyde has pleasant almond, nutty and stony fruit aromas of peach (Prunus persica)” to “Benzaldehyde has pleasant almond, nutty and stony fruit like aromas of peach (Prunus persica)” on the lines 277–278 in Discussion.

---

## [Editor Report · Decision Letter 1]

30 Mar 2020

Odor-active compounds from the gonads of Mesocentrotus nudus sea urchins fed Saccharina japonica kelp

PONE-D-20-02508R1

Dear Dr. Yukio Agatsuma,

We are pleased to inform you that your manuscript has been judged scientifically suitable for publication and will be formally accepted for publication once it complies with all outstanding technical requirements.

With kind regards,

Filippo Giarratana

Academic Editor

PLOS ONE

---

## [Editor Report · Acceptance letter]

2 Apr 2020

PONE-D-20-02508R1 

Odor-active compounds from the gonads of *Mesocentrotus nudus* sea urchins fed *Saccharina japonica* kelp 

Dear Dr. Agatsuma:

I am pleased to inform you that your manuscript has been deemed suitable for publication in PLOS ONE. Congratulations! Your manuscript is now with our production department. 

With kind regards,

on behalf of

Dr. Filippo Giarratana 

Academic Editor

PLOS ONE